# Carbon Footprint of a Large Yellow Croaker Mariculture Models Based on Life-Cycle Assessment

**Jingyi Liu** [1], **Feng Gui** [2], **Qian Zhou** [2], **Huiwen Cai** [2], **Kaida Xu** [3] **and Sheng Zhao** [2,*]

[1] National Engineering Research Center for Marine Aquaculture, Zhejiang Ocean University, Zhoushan 316022, China; liujingyi@zjou.edu.cn
[2] Marine Science and Technology College, Zhejiang Ocean University, Zhoushan 316022, China; fgui@zjou.edu.cn (F.G.); zhouqian@zjou.edu.cn (Q.Z.); caihuiwen@zjou.edu.cn (H.C.)
[3] Zhejiang Province Key Laboratory of Mariculture and Enhancement, Zhejiang Marine Fisheries Research Institute, Zhoushan 316021, China
[*] Correspondence: zhaosh@zjou.edu.cn

**Abstract:** According to the life-cycle assessment method, in this study, we took the traditional plate cage (TPC) mariculture and deep water wind wave-resistant cage (DWWWRC) mariculture of large yellow croaker in China as the research object. We counted and calculated the carbon footprint of the whole life cycle of large yellow croaker cultured in Zhoushan. By analyzing and comparing the advantages and disadvantages of the two according to a perspective of carbon emissions, we found that the carbon footprint of DWWWRC was smaller than that of TPC, which is more suitable for China's large yellow croaker mariculture. We proposed corresponding measures to reduce carbon emissions, such as using clean energy, extending cage life, and improving feed utilization. This study fills the gap in the current research direction of the carbon footprint of large yellow croaker farming in China and provides strong technical support for the sustainable development of China's large yellow croaker cage farming industry.

**Keywords:** life-cycle assessment; large yellow croaker; cage mariculture; carbon footprint; sustainable environment





## 1. Introduction

Aquaculture is the fastest-growing animal production industry in the world, and it is widely considered to be an important way to reduce the widening gap between fish demand and supply [1]. As a result, it has been deeply integrated into the global food system [2]. Mariculture, or the farming of brackish and marine species, accounts for one-third of total aquaculture production and has received increasing attention as a possible supplement to wild marine fisheries [3]. Mariculture accounts for 37.5% of this production and 97% of the global seaweed harvest. To improve livelihoods, its growth has been fostered in many countries, with varying degrees of success [4]. It has been suggested, however, that this expansion of mariculture will lead to increased environmental problems, such as eutrophication, water pollution, threats to biodiversity, and increased greenhouse gas (GHG) emissions [5]. It is well known that the most serious problem of the 21st century is that GHG emissions lead to climate warming. Therefore, governance methods, such as environment and politics, have been proposed to solve this problem [6].

The concept of a carbon footprint evolved from the concept of an ecological footprint, which is used to assess the consumption of natural resources by certain activities [7]. The carbon footprint is used to evaluate the total GHGs released during the entire life cycle of certain products and services, from the extraction of raw materials, manufacturing, assembly, and transport to product use, disposal, and waste management [8]. The carbon footprint is defined as the carbon emissions released by human activities and is calculated as tons of carbon dioxide ($CO_2$) equivalent each year [9]. As many countries and organizations

have developed and published carbon footprint accounting standards for different system levels, many types of carbon footprint standards are currently in use.

GHG emissions continue to increase annually, causing global temperatures to rise. At the national, departmental, or regional level, the most commonly used international standards are the Intergovernmental Panel on Climate Change (IPCC) Guidelines for National Greenhouse Gas Inventories 2006 and the ICLEI (Local Governments for Sustainability) Guidelines for Urban Greenhouse Gas Inventories 2009. Research on carbon footprints has been conducted in a variety of fields and areas around the world, including at the scales of countries and economies [10,11], cities and regions [12–15], households [16], products [17,18], and agriculture and aquaculture [19–21]. After many years of development, the assessment criteria for product carbon footprints are now largely based on life-cycle assessment (LCA). LCA, which involves the compilation and evaluation of inputs, outputs, and potential environmental impacts over the life cycle of a product system, is used worldwide to assess the environmental impacts of industrial production and helps companies and governments to propose improvements [22]. LCA has been recommended by the European Commission (EC) as the best framework for product assessment [23].

LCA is a useful technique that is often employed to quantitatively assess the environmental impact of products, processes, or technologies across the various different stages of their life cycles [24]. In general, the life cycle of a product or technology refers to the entire period of its existence, from the cradle to the grave, which covers the manufacturing, usage, maintenance, and final disposal stages [25]. Among the available multi-impact assessment tools, focusing on quantifiable emissions and energy flows in production processes, LCA methodologically represents one of the most advanced tools, which is increasingly used in aquaculture studies [26]. Over the past 16 years, LCA has been applied in the aquaculture sector to rearing technologies [27,28]; to different species, such as salmon [29,30], rainbow trout [31,32], striped catfish [33], tilapia [27,34], and shrimp [35]; and to aquaculture feeds [36,37]. Relevant LCA studies related to species of Mediterranean origin, such as gilthead seabream [38], seabass [39], and mussel [40], have also been increasingly published, reflecting both the significance of the sector and the usefulness of the tool.

China is the world's leading producer of food from mariculture, accounting for 48% and 62% of global production in 2008 and 2016, respectively [41]. The proportion of food from mariculture in the diet is gradually increasing, but few analyses have examined the carbon footprint of these industries in China [42]. By 2050, the amount of food produced from mariculture is expected to meet 5–19% of the estimated increase in total protein demand of 9.7 billion people worldwide [43,44]. China's mariculture has developed rapidly, which has not only improved the nutritional structure of the Chinese people, enhanced the prosperity of the rural economy, and ensured food security, but also played a positive role in achieving peak carbon neutrality [45]. Therefore, mariculture must grow sustainably to increase fish production for a growing global population, while immediate action to reduce GHG emissions remains a priority for the mariculture industry worldwide [46].

Fish culture in cages dates back many centuries in China [47]. Recently, this practice has spread globally because of its advantages. Cage culture has many advantages over other methods of fish farming, including the very high production per unit volume of water; relatively low investment per unit of production; expected high levels of viability; the use of existing water bodies, reducing pressure on land; the need for relatively low capital investment; ease of movement and relocation; reduced impact of drought on production in relation to water availability; and flexibility of management [48,49].

China's aquaculture industry has grown dramatically in recent years and currently accounts for 60.5% of global aquaculture production [50]. Cage mariculture is a major aquaculture practice in China. The total production of mariculture fish in China reached 21.3531 million tons, of which the production of caged fish was 858,200 tons, accounting for 4.02% of the national mariculture production [51]. Large yellow croaker has a long history of cage aquaculture, and the economic benefits of its aquaculture are significant. In 2020, farmed large yellow croaker production increased to 254,100 tons, accounting for 0.39% of

total marine aquaculture production [52]. In this study, we used the case study method to compare traditional plate cage (TPC) mariculture and the deep water wind wave-resistant cage (DWWWRC). We used the LCA system to compare the carbon footprint of 1000 kg of large yellow croakers according to these two farming modes. From the perspective of carbon emission, we determined the influencing factors of high carbon emission in the two farming modes and proposed the corresponding energy-saving and emission reduction measures. This study provides a scientific reference for the sustainable development of the aquaculture industry.

Section 2 of this paper discusses materials and methods, mainly describing the application of LCA in rhubarb netting; Section 3 features the calculation results, calculating the carbon footprint of two types of netting based on relevant data; Section 4 discusses the results; and Section 5 provides conclusions.

## 2. Material and Method

### 2.1. Research Objectives and Scope Definition

The yellow croaker is an endemic marine fish unique to China and has played a pivotal role in the structure of China's marine fish system. Today, the two main types of large yellow croaker culture in China are (1) TPC mariculture, which is also the most common mode of large yellow croaker culture; and (2) DWWWRC mariculture. We selected these two types of cage mariculture systems of large yellow croaker in Zhoushan City, Zhejiang Province, as the carbon footprint research object. The purpose of this research was to quantify and compare the carbon footprint caused by resource and energy consumption in the whole life cycle of the two cage mariculture systems. We defined the scope of this study as the process from fry (semi-adult) to formation to adult of large yellow croaker. This study included the production of materials (electricity, diesel, wood, polyethylene (PE), compound feed, formaldehyde, refrigerators, and cement) required for the construction of the cage structures and the daily operation of the large yellow croaker, the transportation and rearing of various materials, energy consumption, and $CO_2$ emissions. It did not include the transport of large yellow croaker to market, sales, or consumer consumption.

### 2.2. Study Area

In China's marine cage aquaculture industry, the most typical fish farmed is the large yellow croaker. The sea area of Zhoushan Islands is one of the main production areas of large yellow croaker. Zhoushan large yellow croaker is the first example of geographically trademarked seafood in China, and it is the only one to date. The large yellow croaker produced in Zhoushan is fresh and nutritious and is much loved by the Chinese people. The study area was the Dongji Islands, Zhoushan City, Zhejiang Province (Figure 1). We obtained the data from Zhejiang Big Ocean Technology Co., Ltd., which is a breeding company with a high market share in the Zhoushan large yellow croaker mariculture industry and has carried out TPC and DWWWRC breeding of large yellow croaker in the Dongji Island area. The DWWWRCs in this study were circular floating cages made of high-density PE with a circumference of 40 m. The cages consisted of a full-floating gravity cage frame, breeding net, and fixed system. The TPCs in this study were small panel floating cages. The cage specification was 4 m × 4 m × 4 m and consisted of board, netting, and a floating dust device.

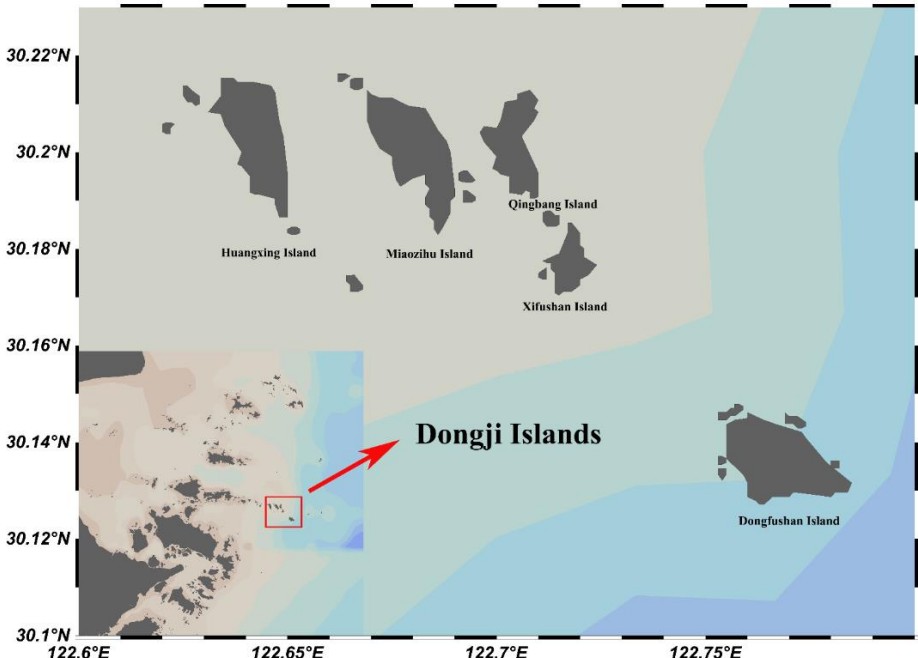

**Figure 1.** Schematic diagram of the Dongji Islands in Zhoushan.

We purchased the seedlings of large yellow croaker from Fujian Province and placed them in a mariculture base for artificial culture. Each cage initially had about 4000 juveniles, and bait was fed twice a day. The mariculture period was about 6 months, and the survival rate was about 95%. The harvest yield was 35.75 tons, and the average individual weight in December was 550 g. The average market price of large yellow croaker was 100 yuan/kg.

### 2.3. Large Yellow Croaker Cage Mariculture

The basic design of a mariculture system consists of cages and working/fishing vessels with various equipment. These devices are used for breeding, feeding, cooling, and storage, which will consume energy and release greenhouse gases [53].

The DWWWRC was made of a high-density PE circular floating cage with a circumference of 40 m. It was composed of a fully floating gravity cage frame, breeding mesh, and fixing system, as shown in Figure 2a. The main frame of the DWWWRC was made of high-density PE material, which included three or four circular tubes and brackets. The bottom ring was made of two or three 250-mm diameter pipes, and the upper and lower pipes were linked by PE brackets. The support frame of a circular floating frame cage had good distortion and high strength. The net coat of a DWWWRC generally was made of PE material or nylon material (polyamide). The material of the net coat effectively protected the big yellow croaker and prevented the big yellow croaker from scratching. The bottom of the cage was given a certain amount of weight to ensure that the net maintained its form underwater.

The TPC was a small plate-type floating cage. The size of the cage was 4 m × 4 m × 4 m. The cage consisted mainly of wood, PE netting, and a dust-floating device, as shown in Figure 2b. The TPC is a combined wooden structure cage. Considering the instability of its structure, this type of cage is usually placed in the inner bay of the ocean, and therefore, it is also called the inner bay cage. In Southeast Asia, this inner bay-type cage has a flat wooden structure. In the past, the buoyancy of the cage was provided by a cylindrical foam float. The float was tied to the bottom of the frame board with a 120-strand PE rope so that the frame was approximately 30 cm above the sea. On the outside of the traditional wooden cage frame, two or three PE floats were mounted 1–1.5 m apart. To prevent the netting from deforming under the action of the tide, we added weights around the bottom

of the netting to hold it in place. The number of cement blocks used depended on the size of the box.

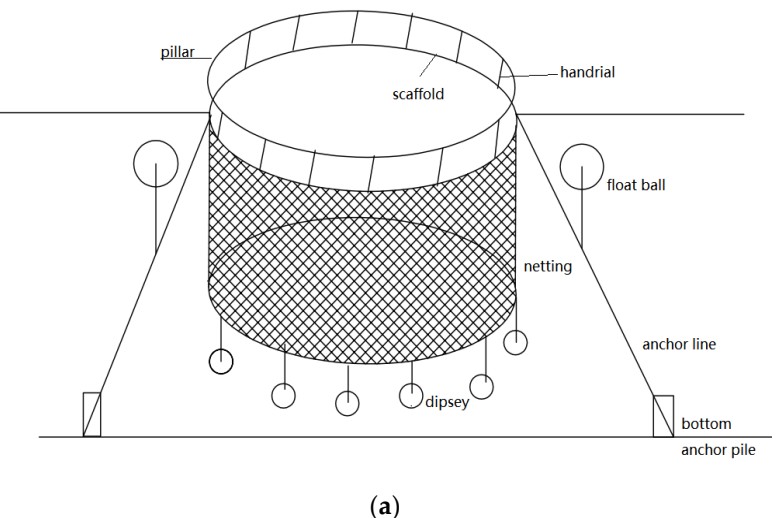

(**a**)

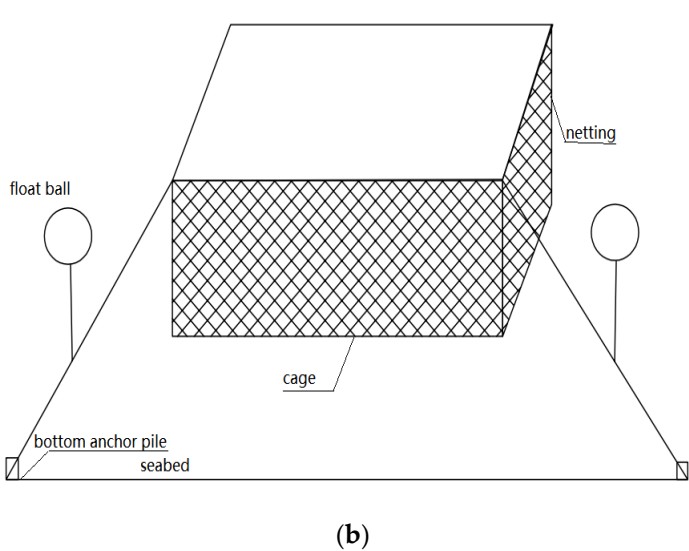

(**b**)

**Figure 2.** (**a**) Schematic diagram of the DWWWRC. (**b**) Schematic diagram of the TPC.

### 2.4. Division of System Boundaries and Functional Units in LCA

The LCA method includes the determination of a research object and purpose, the definition of research scope, inventory analysis, carbon emission calculation, and carbon footprint analysis. The LCA for aquaculture production systems is focused on an assessment of the environmental effects of producing farmed fish, considering all inputs, resources, and waste corresponding to the entire production cycle [26].

### 2.5. System Boundaries and Functional Units

Figure 3 shows a dashed box, which is the LCA system's boundary of large yellow croaker mariculture in a cage. We selected the cradle-to-farm gate as the cutoff system boundary in the LCA, without considering the carbon emissions generated by consumers in the process. In this case, the LCA began in the cage (e.g., materials, electricity, diesel, formaldehyde, PE, refrigerators, cement, and transportation) and included the feed, fry input, and energy demand required to produce tons of cultured large yellow croaker.

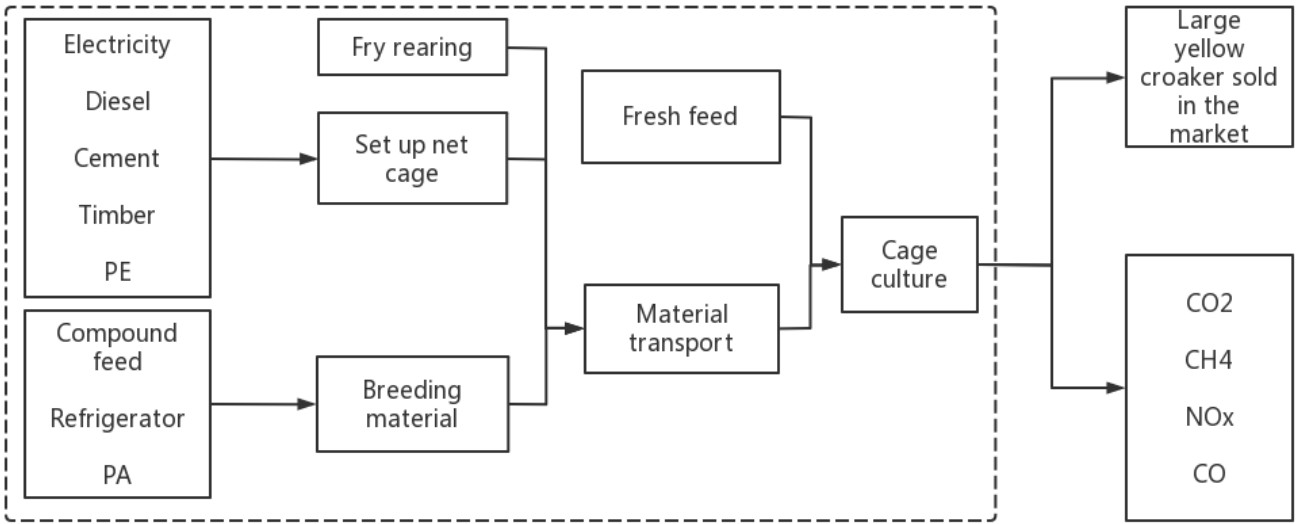

**Figure 3.** LCA system's boundary of large yellow croaker mariculture in cage.

*2.6. Data Collection*

We contacted the representative large yellow croaker farmers of Zhejiang Big Ocean Science and Technology Co., Ltd. to conduct a face-to-face questionnaire survey in 2021. We randomly selected several farmers of Dongji large yellow croaker to collect relevant data. We then collected the farmers' production information for the basic situation of large yellow croaker breeding, feed use, cost input, and breeding income. All data collection and summary of information was done by team members. Informed consent forms were given to the farmers before they completed the survey. Because the farmers did not understand the carbon footprint and LCA method, we communicated with them face-to-face so that we could use the LCA to build a bridge between the carbon footprint and farming practices and to collect data to more accurately calculate the GHG emissions of farmed large yellow croaker. After completing the field survey, we obtained the relevant farming data from the farmers, as shown in Table 1. According to the general service life of building materials, we identified details on the following: construction of large DWWWRC and small TPC; consumption of building materials and energy according to a service life of 15 years; high-density PE plastic frame pipe according to a service life of 12 years; PE netting according to a service life of 2.5 years; refrigeration according to a service life of 10 years; and anticorrosion wood according to a service life of 1 year, which was converted to 0.5 years of service life. See Table 1 for details.

**Table 1.** Investment table for breeding 1000 kg of large yellow croakers.

| Material Input | DWWWRC | TPC |
| --- | --- | --- |
| Diesel/kg | 46.21 | 310.69 |
| Electricity/(kW·h) | 13.53 | 17.41 |
| PE/kg | 64.45 | 72.61 |
| Cement/kg | 1.80 | 6.31 |
| Compound feed/kg | 336 | 1736 |
| Fresh feed/kg | 3364 | 6844 |
| PA/kg | 0.079 | 8.45 |
| Timber/$m^3$ | 0 | 13.19 |
| Refrigerator/unit | 1 | 1 |

*2.7. Computational Process*

According to the LCA calculation method, we calculated the life cycle of the large yellow croaker cage mariculture system, defined the research objectives and scope, and carried out the stock analysis, impact assessment, and interpretation of the results.

Research objective and scope: We calculated the carbon emissions of cage culture and raft culture for large yellow croaker and adopted the corresponding emission reduction measures.

Inventory analysis: We collected data on the consumption of resources and energy for the two large yellow croaker cage culture systems, determined their emission factors, and then calculated the carbon emissions for certain levels of resources and energy. The unit was $CO_2$-eq, and the calculation Formula (1) was as follows [45]:

$$Ci = Vi \times Fi \tag{1}$$

where $Ci$ is the carbon emissions of the $i$th energy or substance, $Vi$ is the consumption of the $i$th resource or energy, and $Fi$ is the emission factor of the $i$th resource or energy. This is the coefficient corresponding to the data at the activity level, including the carbon content per unit calorific value or elemental carbon content and the oxidation rate, and the GHG emission coefficient representing the unit production or consumption activity. Thus, the total carbon emissions during the life cycle of cage mariculture of large yellow croaker can be expressed as follows in Formula (2):

$$C = \sum_{i=1}^{n} Ci \tag{2}$$

where $C$ is the total carbon emissions and $Ci$ represents the carbon emissions of the $i$th energy or substance.

### 2.8. Data Processing

The carbon emissions of each material can be calculated using Formula (1). The emission factors can be calculated either directly from known data (i.e., the default value) provided by the IPCC, the U.S. Environmental Protection Agency (EPA), or the European Environment Agency, or from representative measured data.

In this study, the carbon emission data for the energy production and use process, PE, refrigerators, preservative wood, cement, and formaldehyde refer to Di [54]; Hu [55]; Li [56]; Xiao [57]; Shen [58]; Nakano [59]; and Xiao [60]. According to Brentrup's [19,20] research, the carbon emission coefficient of compound feed is 0.26 kg $CO_2$ per kilogram of feed. The contents of $CO_2$, carbon monoxide (CO), methane ($CH_4$), and nitrogen oxide (NOX) in GHG are relatively large. We considered only these four gas emissions. By collecting the carbon emissions of the two cage mariculture systems (Table 2), we determined the emission factor and then calculated the carbon emission of a certain resource or energy according to Formula (1).

**Table 2.** Carbon emissions of various materials.

| Classification | PE | Cement | Timber | PA | Diesel | Refrigerator |
|---|---|---|---|---|---|---|
| | 1T | 1T | 1 m$^3$ | 1T (37%) | 1 kg | 1 |
| CO2/kg | $9.11 \times 10^1$ | $6.98 \times 10^2$ | $1.45 \times 10^2$ | $1.69 \times 10^2$ | $6.07 \times 10^{-1}$ | $2.66 \times 10^3$ |
| CO/kg | $2.64 \times 10^0$ | $1.26 \times 10^{-1}$ | $1.36 \times 10^{-1}$ | $9.77 \times 10^0$ | $3.88 \times 10^{-4}$ | $1.01 \times 10^0$ |
| CH4/kg | $0.00 \times 10^0$ | $8.97 \times 10^{-1}$ | $1.36 \times 10^1$ | $3.78 \times 10^0$ | $1.18 \times 10^{-4}$ | $1.15 \times 10^1$ |
| NOx/kg | $1.03 \times 10^0$ | $1.38 \times 10^0$ | $3.65 \times 10^{-1}$ | $0.00 \times 10^0$ | $6.71 \times 10^{-1}$ | $7.19 \times 10^0$ |

## 3. Results

In this study, we used a 5-ton truck using diesel fuel for transport. For the emissions of $CO_2$ and other GHG pollutants from diesel combustion, we referred to the EPA's list of diesel production and combustion for automobiles [57]. According to the weight of the goods and the transportation distance, the transportation volume could be calculated. For transport distance, we used the driving distance from the shortest route of self-driving cars

on Google Maps. The fuel consumption of trucks with 5 tons of diesel fuel was 0.167 kg of diesel. Using this value, we calculated the fuel consumption of various materials during transportation. The details are given in Table 3. Using $CO_2$ as a reference for global warming [61], the equivalent coefficients of GHGs including $CO_2$, CO, $CH_4$, hydrocarbon (HC), and $NO_X$ were 1, 2, 25, 25, and 320 $kgCO_2$-eq·$t^{-1}$, respectively. According to the calculation Formula (2), we obtained the carbon footprint per unit of large yellow croaker production in the two cage mariculture systems by adding the product of the resource and energy consumption given in Table 1 and the carbon emission coefficient given in the literature, as shown in Table 4.

**Table 3.** The diesel consumption in transportation of cages' building materials and breeding stage.

| Material Name | Weight (kg) | | Distance (km) | | Oil Consumption (kg) | |
|---|---|---|---|---|---|---|
| | TPC | DWWWRC | TPC | DWWWRC | TPC | DWWWRC |
| Timber | $6.20 \times 10^3$ | 0 | $8.94 \times 10^1$ | 0 | $9.26 \times 10^1$ | 0 |
| Compound feed | $1.74 \times 10^3$ | $3.36 \times 10^2$ | $7.29 \times 10^2$ | $7.29 \times 10^2$ | $2.12 \times 10^2$ | $4.09 \times 10^1$ |
| Cement | $6.30 \times 10^0$ | $1.80 \times 10^0$ | $2.30 \times 10^2$ | $2.30 \times 10^2$ | $2.42 \times 10^{-1}$ | $6.91 \times 10^{-2}$ |
| Refrigerator | $5.00 \times 10^1$ | $5.00 \times 10^1$ | $2.30 \times 10^2$ | $2.30 \times 10^2$ | $1.92 \times 10^0$ | $1.92 \times 10^0$ |
| PE | $7.26 \times 10^1$ | $6.45 \times 10^1$ | $4.11 \times 10^2$ | $4.11 \times 10^2$ | $4.98 \times 10^0$ | $4.43 \times 10^0$ |
| PA | $8.48 \times 10^0$ | $7.86 \times 10^{-2}$ | $2.88 \times 10^2$ | $2.88 \times 10^2$ | $4.08 \times 10^{-1}$ | $3.78 \times 10^{-3}$ |

**Table 4.** $CO_2$ emissions per unit of yield (1000 kg) in two types of large yellow croaker cage mariculture systems.

| Name | Unit | DWWWRC | TPC |
|---|---|---|---|
| Diesel | $kgCO_2$-eq | $9.95 \times 10^3$ | $6.69 \times 10^4$ |
| Electricity | $kgCO_2$-eq | $4.34 \times 10^1$ | $5.56 \times 10^1$ |
| PE | $kgCO_2$-eq | $3.89 \times 10^1$ | $4.38 \times 10^1$ |
| Cement | $kgCO_2$-eq | $2.09 \times 10^0$ | $7.32 \times 10^0$ |
| Compound feed | $kgCO_2$-eq | $8.74 \times 10^1$ | $4.51 \times 10^2$ |
| PA | $kgCO_2$-eq | $7.83 \times 10^{-2}$ | $6.46 \times 10^0$ |
| Timber | $kgCO_2$-eq | $0.00 \times 10^0$ | $7.93 \times 10^3$ |
| Refrigerator | $kgCO_2$-eq | $4.33 \times 10^2$ | $1.37 \times 10^2$ |
| Total $CO_2$ emissions | $kgCO_2$-eq | $1.055 \times 10^4$ | $7.553 \times 10^4$ |

As shown in Table 4, we compared the two cage mariculture systems for 1000 kg of large yellow croakers. We found that the total $CO_2$ emission of the DWWWRC was $1.055 \times 10^4$ $kgCO_2$-eq·$t^{-1}$ and that of the TPC was $7.553 \times 10^4$ $kgCO_2$-eq·$t^{-1}$, and the ratio between the two was approximately 1:7.16.

According to Table 4, in the TPC mariculture system, the carbon emission caused by diesel combustion consumption ranked first, which was as high as $6.69 \times 104$ $kgCO_2$-eq·$t^{-1}$. Diesel combustion was the main source of carbon emission, accounting for 88.03%. Wood accounted for 10.43% of the total emissions, with a total emission of $0.793 \times 104$ $kgCO_2$-eq·$t^{-1}$, and was another major source of carbon emissions. The TPCs were constructed mainly of wood, which was easily corroded because of the influence of seawater salinity and increased the input cost. In addition, the processing and replacement of wood also increased GHG emissions.

In the DWWWRC mariculture system, the carbon emission caused by diesel oil consumption was $0.995 \times 10^4$ $kgCO_2$-eq·$t^{-1}$, which was less than that of the TPC mariculture model, but this was still the main cause of carbon emission in the system, accounting for 94.31%. Diesel consumption was mainly the result of transporting materials, feeding large yellow croakers, and other processes. The $CO_2$ and NOx emissions from diesel combustion were the main causes of GHG emissions. In this study, the emission and equivalent factor of NOx was high, and the $CO_2$ equivalent value accounted for 77.65% and 99.67% of the total amount in the transport stage, respectively. In the process of production and use, the GHG

emitted by the refrigerator was $4.33 \times 10^2$ kgCO$_2$-eq·t$^{-1}$, accounting for 4.1% of the total emissions, which was a minor carbon emission source. Figure 4 compares CO$_2$ emissions.

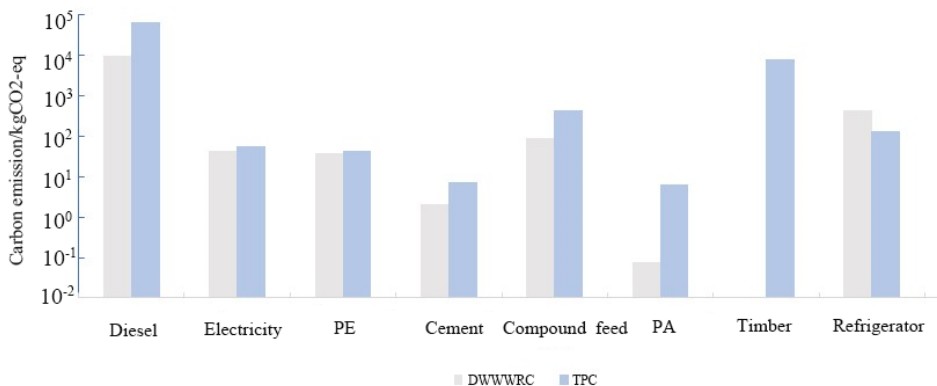

**Figure 4.** CO$_2$ emissions per unit of yield (1000 kg) in two types of large yellow croaker cage mariculture models.

GHG emissions are caused mainly by the transport phase of cage mariculture materials. In the transport stage of the two cage mariculture systems, the transport oil consumption of compound feed was the largest contributor to these emissions, accounting for 86.42% and 67.9% of DWWWRC mariculture and TPC large yellow croaker cage mariculture, respectively (see Table 2). At this stage, the GHG emitted by DWWWRCs and TPCs was $0.995 \times 10^4$ kgCO$_2$-eq·t$^{-1}$ and $6.69 \times 10^4$ kgCO$_2$-eq·t$^{-1}$, respectively, accounting for 94.30% and 88.57% of the total, as shown in Table 5. The research area in this study is located in the northeast sea area of Zhejiang Province, which is far from the cities and provinces of China and undoubtedly increased diesel consumption. The carbon footprint of the DWWWRC mariculture model was $8.44 \times 10^1$ kgCO$_2$-eq·t$^{-1}$ in the production stage, $5.20 \times 10^2$ kgCO$_2$-eq·t$^{-1}$ in the breeding stage, and $9.95 \times 10^3$ kgCO$_2$-eq·t$^{-1}$ in the transportation stage. The carbon footprint of the production stage of the TPC mariculture model was $8.04 \times 10^3$ kgCO$_2$-eq·t$^{-1}$, that of the breeding stage was $5.94 \times 10^2$ kgCO$_2$-eq·t$^{-1}$, and that of the transportation stage was $6.69 \times 10^4$ kgCO$_2$-eq·t$^{-1}$. Under the two cage mariculture systems, the carbon footprint of each stage is shown in Table 5.

**Table 5.** CO$_2$ emissions from two kinds of cage mariculture in different processes.

| Process | DWWWRC | | TPC | |
|---|---|---|---|---|
| Material production | $8.44 \times 10^1$ | 0.80% | $8.04 \times 10^3$ | 10.64% |
| Breeding process | $5.20 \times 10^2$ | 4.90% | $5.94 \times 10^2$ | 0.79% |
| Material transport | $9.95 \times 10^3$ | 94.30% | $6.69 \times 10^4$ | 88.57% |

## 4. Discussion

In this study, we provided relevant data for two farming modes, TPCs and DWWWRCs, and calculated the life cycle of these two mariculture models. The carbon footprint of these two farming modes can largely represent the carbon emissions of the large yellow croaker cage mariculture industry in Zhoushan City. To gain a more comprehensive and in-depth understanding of the carbon footprint of large yellow croaker cage mariculture in Zhoushan City, it is necessary to comprehensively collect and analyze the breeding facilities and various inputs. However, we found deficiencies in the two farming modes. For example, in TPC mariculture, the GHG content of building cages was higher than that of cage mariculture. Both farming modes have disadvantages that require farmers to make changes and pursue innovations in farming technology and management to improve farm profits and the sustainability of the farming modes.

### 4.1. Effect of Mariculture Model on Carbon Footprint

According to the data results of this study, the GHG emissions generated by DWWWRC in the production stage, aquaculture cage mariculture, and transportation stage were lower than those of TPC aquaculture. From the perspective of the carbon footprint, the DWWWRC mariculture model was better than the TPC mariculture model, which aligns with the requirements of sustainable development. Both mariculture models have different problems. For example, the many problems of traditional small-cage mariculture include their small aquaculture capacity, poor ability to withstand wind and waves, short life span, and severe limitation of aquaculture area. These problems have resulted in environmental pollution, fish diseases, poor fish quality, severely limited development of mariculture, and a series of economic, environmental, and social problems. Although the DWWWRC has many advantages, including high strength, good flexibility, corrosion resistance, anti-aging, strong wind and wave resistance, long service life, large effective aquaculture water area, and high efficiency, the initial investment cost is high. Existing cage support facilities need to be further improved and safety further enhanced as climate change undoubtedly will challenge the future growth of marine aquaculture [62].

In this study, we used the LCA method to calculate the carbon footprint, identify the most influential factors of carbon emissions, and propose improvement measures. Through this research, it was evident that the emission reduction measures of cage farming large yellow croaker were to use recyclable and durable materials to build the cages, pay attention to the reasonable and scientific design of the cages, and improve productivity and constantly optimize breeding [63]. Mariculture is a relatively new industry compared with terrestrial animal husbandry. Technological improvements such as breeding and genetic selection can be made to reduce GHG emissions from large yellow croaker farming, and improving reproductive performance is helpful in reducing $CH_4$ emissions [64]. In this study, feed was the main source of GHG, and by implementing automatic feeders and monitoring systems in seawater farms, we could control feed consumption, reduce costs, and reduce GHG emissions in China [65,66]. Another good solution to reduce emissions is to switch energy sources [23]. It is known that hydropower, nuclear power, and wind power are the three major clean energy sources in China. In the future, clean energy can be considered to reduce energy-related GHG emissions [67].

### 4.2. Impact of Emission Factors on Carbon Footprint

The emission factor is the coefficient corresponding to the activity-level data, including carbon content per unit calorific value or elemental carbon content and oxidation rate, which can be used to characterize the GHG emission coefficient per unit of production or consumption activity. Emission factors can be calculated directly from the known data (i.e., default values) provided by the IPCC, the EPA, and European environmental agencies, or can be based on representative measurement data [68]. The emission factors selected for the same emission source in different industries are also different, which will have a corresponding effect on the calculation results.

In this study, the selection of emission factors came from the published literature, and the carbon emission data, including the energy production and use process, PE, refrigerators, preservative wood, cement, and formaldehyde, came from Di [54]; Hu [55]; Li [56]; Xiao [57]; Shen [58]; Nakano [59]; and Xiao [60]. Because of the variety of emission factors, it is necessary to determine the industry and the composition of the activity data when determining the emission factors [65]. Doing so will improve the accuracy of the calculation results.

### 4.3. The Influence of System Boundary on Carbon Footprint

The system boundaries must be considered on the basis of their versatility. Not all process boundaries can be tested and validated. Some processes are not synchronized with the assessment. Henriksson found that aquaculture LCAs often require large system boundaries, including fisheries, agriculture, and livestock production systems from around

the world [69]. In addition to different choices of functional units, system boundaries, and impact assessment methods, studies also differed in the choice of allocation factors and data sources [70–72]. The interpretation of the results also differed between these studies, and a number of methodological choices have been identified that may have influenced the results. In this study, the cradle-to-gate boundary was defined, and market sales were not included in this boundary to more accurately calculate the carbon footprint of large-scale yellow croaker farming.

## 5. Conclusions

In this study, we used the LCA method to calculate the carbon footprint of 1000 kg of large yellow croakers under two farming modes. According to the three stages of production, breeding, and transportation, we examined eight aspects of carbon emissions, including electricity, wood, PE, feed, formaldehyde, refrigerators, cement, and diesel.

The results showed that the total carbon footprint of 1000 kg of large yellow croakers was $1.055 \times 10^4$ kgCO$_2$-eq·t$^{-1}$ and $7.55 \times 10^4$ kgCO$_2$-eq·t$^{-1}$. In addition, diesel consumption was the largest contributor to these emissions, accounting for 94.31% and 88.03%. In the DWWWRC mariculture model, diesel was the first emission source, accounting for 94.31%; refrigerators were the second emission source, accounting for 4.1%; and formaldehyde was the lowest emission source. In the TPC mariculture model, diesel was the first emission source and accounted for 88.03%; wood was the second emission source and accounted for 10.5%; feed, refrigerators, electricity, and PE followed in turn, and formaldehyde was the lowest emission source.

From the results of this study, the carbon emissions of the DWWWRC system in terms of capital construction materials, feed, and energy were much lower than those of the TPC system. That is, from the point of view of carbon footprint, the DWWWRC mariculture model was clearly better than the TPC mariculture model, which aligned with the requirements of sustainable development.

In this study, we discussed the limitations of the calculation results from three influencing factors: breeding mode, emission factor, and system boundary. At the same time, we proposed measures to reduce carbon footprint emissions, such as using clean energy, increasing the service life of cages, and improving feed utilization. This study can help marine farmers choose a certain cage culture to reduce carbon emissions and can promote the whole large yellow croaker aquaculture industry to achieve the goal of a sustainable environment as soon as possible.

Finally, although DWWWRC mariculture still has shortcomings, with continued progress in science and technology and the replacement of materials, new types of cages with energy savings and emissions reduction are certain to be produced in the future. At the same time, we should continue to learn and better understand the basic knowledge of mariculture management methods and nutrient emissions to fully realize the sustainable development of cage aquaculture as soon as possible.

**Author Contributions:** J.L.: Conceptualization, Data curation, Methodology, Writing—original draft, Formal analysis, Investigation. F.G.: Conceptualization, Supervision. Q.Z.: Investigation. H.C.: Supervision, Methodology. K.X.: Supervision. S.Z.: Conceptualization, Supervision, Methodology, Writing, Funding acquisition. All authors have read and agreed to the published version of the manuscript.

**Funding:** This study is sponsored by the National Key R&D Program of China (No. 2019YFD0901204); the Fundamental Research Funds for Zhejiang Provincial Universities and Research Institutes (No. 2021JD006); and the Key R&D Program of Zhejiang Province (No. 2019C02056).

**Institutional Review Board Statement:** Not applicable.

**Informed Consent Statement:** Not applicable.

**Data Availability Statement:** Not applicable.

**Acknowledgments:** We thank LetPub (www.letpub.com) for its linguistic assistance during the preparation of this manuscript.

**Conflicts of Interest:** The authors declare no conflict of interest.

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
