# Peer review of "Carbon Footprint of a Large Yellow Croaker Mariculture Models Based on Life-Cycle Assessment"

_sustainability, doi:10.3390/su15086658_

Round 1

Reviewer 1 Report

Dear Authors,

your manuscript briefly compare carbon footprint of two farming systems of a large yellow croaker in a selected area. The study is interesting, however, there are several shortcomings. Please, see my thoughts and recommendations below.

1. Consider using acronyms for the farming systems.

2. Do you consider the croaker as an input flow? You mention only compound feed. If not, please explain why it is excluded from the study.

3. How far from the coast and from each other are the farming systems located?

4. You mention the traditional plate cage dimensions are 4x4x4 m and the deep-water wind-wave-resistant cage circumference is 40 m. Can you define how many croakers are able to live in each cage? Moreover, what is the depth of the deep-water wind-wave-resistant cage?

Line 105 - there is two times the same citation written

Line 112 - font size should be adjusted

Figure 3 - the figure seems disarranged. You should consider the design and maybe generate a new figure with a description and precisely defined flows within the system boundary as well as the transport, production and breeding stage. Note: consider merging "phase" and "stage".

Lines 169-170 - you mention cage and raft culture. What is the difference between them?

Lines 216-217 - I think you swapped Figure 1 for Figure 4

Lines 226-232 - I could not find the evidence in the study. In my opinion these statements are not provided by the Figure 4.

Figure 3 - the unit of the graph is missing

Lines 233-237 - where does this information come from?

Line 239 - you probably swapped Figure 3 for Figure 4

Lines 242-247 - I would recommend showing the values in the figures or graphs, it would be easier to read.

Lines 286-306 - it seems like a literature overview. I would rather move it to the introduction.

Line 310 - you mention recyclable materials for the first time in your study. Which materials you consider recyclable?

Line 311 - can you define how the breeding can be optimised? What was the difference in the feeding in both farming systems?

Line 314 - you mention feed was the main source of GHG. Can you explain the statement? In the results, there are more polluting elements.

Line 323 - please, support your statement with a citation

Line 325 - you did only truncated LCA, as you mentioned in the line 139, not a whole LCA.

Line 331 - please, refer to the corresponding farming systems.

Line 340 - you mention "healthy nutrition". I would consider skipping it or adding a supporting citation while your study does not deal with impact of eating fish on human health.

Best,

Reviewer 2 Report

State academic, practical and policy contributions and implications of the paper.

2.1 not research object but research objective.

Line 81, why Zhoushan? Can the results be generalised?

Why data was obtained from from Zhejiang Big Ocean Technology?

Raise resolution of the figures 1, 3. Make sure the words are not distorted. Most figures’ words look very strange now.

Font size of 112 is wrong “deep-sea wind-wave-resistant cage”

Line 154 All of these interviews and face-to-face questionnaires were completed by our team members. I cannot see any interview results, please add that back

Please add the survey questions in the paper.

It is unclear what does LCA use in this paper. Please cite some top papers of LCA A comparative life-cycle assessment of hydro-, nuclear and wind power: A China study - ScienceDirect

The research method is too simple, we cannot use bar charts only for publishing a SSCI paper.

Modelling MUST be added, turn some or all the pie charts to one Table.

Deeper discussion is needed.

State the academic, practical and policy contributions of this paper.

State the limitations of this paper.

As the paper is CO2 related, there should be a section about carbon dioxide included there as well.

Papers like could be cited “A Study on Public Perceptions of Carbon Neutrality in China: has the Idea of ESG Been Encompassed?”

Dissertation should be replaced by top journals.

[37]Zhang, J. P. (2010).Research on refrigerator life cycle environmental impact asses-sment.(Doctoral disser 456 tation, MSc Dissertation, Shanghai Jiao Tong University). 457 [38]Huang Dong-mei, Zhou Pei-guo, Zhang Qi-sheng.(2012). Life cycle assessment of bamboo constructed 458 house. Journal of Beijing Forestry University, 2012, 34(5): 148-152.

Poor papers like the followings should be removed:

Carbon Footprint of Shrimp Pond A quac-ulture Based on the 488 LCA Method.Chinese Journal Of Managemant Science(eds.)(pp.674-678)

[51]Shipton, T. A., & Hasan, M. R. (2013).An overview of the current status of feed management practice 492 s. FAO Fisheries and Aquaculture Technical Paper, (583), 3-20.

Missing some parts in references:

[52]MacLeod, M., Eory, V., Gruère, G., & Lankoski, J. (2015). Cost-effectiveness of greenhouse gas mitigati 494 on measures for agriculture: a literature review

Line 447, Transactions of the CSAE(11),141-146. Full name of CSAE should be stated.

Please standardise all the references.

 Polish English.

Round 2

Reviewer 1 Report

Dear Authors,

your manuscript has undergone considerable editing. However, I have one more recommendation for you - I find tables 3-2, 4 and 5 appropriate to move to the results section.

Best,

Author Response

Dear Editors and Reviewers,

Thank you very much for taking the time to review this manuscript. We appreciate all your generous comments and suggestions! Please find my revisions in the resubmitted document Response to reviewer comments.

We appreciate your comments on the manuscript. In response to your suggestions we have revised the relevant parts of the manuscript and all your questions have been answered one by one.

Point 1: Your manuscript has been heavily edited. However, I have one more suggestion for you - I Found Form 1-3, 2 and 4 Suitable for moving to the Results Section

Response 1: Thank you very much for the reviewer's suggestion. For ease of reading, we move Tables 4 and 5 to the results section. Thank you for your valuable revisions We have revised the manuscript accordingly, and our point-by-point response is shown in the attachment.

Thank you for your careful review We appreciate your efforts in reviewing our manuscript. These comments are very valuable and helpful to the revision and improvement of our paper. We wish you good health for your family and the community.

Yours sincerely,

Jingyi Liu

Reviewer 2 Report

Copy edit the whole paper, there are a lot of English problems.

Avoid abbreviations in abstract.

Abstract does not need to state figure like this 1.055×104 kgCO2-eq. It only states what are the main findings, like very large and small etc and what is the implication based on the results obtained.

Line 13, remove (TPC) and (DWC)

Line 16, remove (CO2)

Line 17, that of traditional plate cage, compare to what?

Abstract needs rewriting. State the research gap, originality of the research, practical, policy and academic implications of the paper.

The abstract is about mariculture but the title fails to reflect that. Please change the title.

Line 26, Aquaculture instead of Mariculture was used. Are they the same?

Line 34, As a matter of common knowledge, not a proper English writing.

Line 68, research object should be research objective

Lines 68-9, The research object of this study was not the same as that produced in Mexico, but the operating conditions were similar.

Lines 1-55, too long, please split that to two pargraphs.

Lines 56-75, this part needs rewriting.

Figure 1, the words are distorted, a higher resolution figure is needed.

Lines 137-147, too many abbreviations make the whole part difficult to read.

Figure 2 and others need citations.

Figure 3, words need to be clearer.

Line 184, face-to-face research???

2.7 should be data processing

Line 225, Di and Hu. [42;43] wrong citation?

Tables 3.1 and 3.2, Tables 4 and Tables 5 need more descriptions about where do the data come from and they need citations for the sources of data as well.

cage mariculture systems need more description at the beginning of the paper.

247-285 need citations

E+x in the figure looks very strange, we usually only label the data in y-axis, please remove the numbers on top of each bars.

In the y-axis, state the unit like (106) rather than 1.00E+05

Why mariculture industry in Zhoushan City? Can the results be generalised?

4.2 needs an extension based on the previous publications.

Conclusion part needs to state academic, practical and policy contributions, limitations of the research, research gap that it tries to fill etc. it is too short now.

Section 2.2 why the areas were selected? Based on what criteria?

Round 3

Reviewer 2 Report

Ratio like 7.16 : 1 in abstract may not be useful to reader, it may be better to state what is that important if we know that is 7.16 to 1.

Add academic, practical and policy contributions of the paper, originality, research gap that it fills to abstract.

At the end of the introduction, introduces the later sections, e.g. section 2 isxx, section 3 showcases…

2.1, after stating “We selected two types of cage mariculture systems of large yellow croaker in 115 Zhoushan City, Zhejiang Province”, please tell what are these two types. Why are they selected?

Figure 1’s words on the four sides are not very clear, higher resolution is needed. And please add citations.

Check if Figure 1 needs to apply for copyright to put that in the paper. Similar issue for Figures 2a and b for the above conditions.

Figure 3, it is unclear what the figures would like to tell, e.g. CO2 and marketable fish are put to the same rectangular box. What does that mean? Figures should be posted to let the readers know what it is without referring much to the paragraphs.

Zhejiang Big Ocean companies’ materials might need to declare conflict of interest.

Abbreviations like DWCs should be avoided, full name is preferred.

Instead of stating a whole trunk of literatures, Line 357, 19,20,41,50-58, please tell what do these literatures said.

A separate section for LCA method and mariculture is needed, cite some more literature for it: Wang et al (2019) A comparative life-cycle assessment of hydro-, nuclear and wind power: A China study.

I cannot see how LCA is used in the life cycle of mariculture and this has to be highlighted.

Reference 24, (Master) thesis should be avoided.

Table 3.1, how to measure materials’ emission? Please state.

Tables 3.1 and 3.2 need citation.

The results part need to state clearly how the numerical figures are obtained and calculated.

Table 4 needs citation and DWC should be in full name.

A clearer, higher resolution figure for Figure 4 is needed.

It said the research used LCA to calculate, but what is the formula? What are the software involved. What are the stages include in this life cycle of mariculture? Readers cannot see any clear stages in the paper, so how can that be estimated as a life cycle analysis? A figure that demonstrates different stages in mariculture of the two types of cages are needed to see what are the carbon emissions emitted in each stage so that we can know what are the carbon emissions.

48 reference, (05),297 is in wrong format?

Reference 50 has something wrong/missing?
